# Additive Insecticidal Effects of Chitosan/dsRNA Nanoparticles Targeting *V-ATPaseD* and Emamectin Benzoate–Lufenuron Formulations Against *Spodoptera frugiperda* (J.E. Smith) (Lepidoptera: Noctuidae)

**DOI:** 10.3390/insects16040348

**Published:** 2025-03-27

**Authors:** Shigang Guo, Zhongwei Li, Xuhui Zhao, Donghai Zhang, Camilo Ayra-Pardo, Yunchao Kan, Dandan Li

**Affiliations:** 1Henan Key Laboratory of Insect Biology in Funiu Mountain, The International Joint Laboratory of Insect Biology in Henan Province, College of Life Science, Nanyang Normal University, 1638 Wolong Road, Nanyang 473061, China; 19513075927@139.com (S.G.); lzw17337796813@163.com (Z.L.); 17354620935@163.com (X.Z.); 2Xifu Nong Biotechnology Co., Ltd. of Henan, Kaifeng 475500, China; 3CIIMAR/CIMAR LA, Interdisciplinary Centre of Marine and Environmental Research, University of Porto, Terminal de Cruzeiros do Porto de Leixões, Avda. General Norton de Matos s/n, 4450-208 Matosinhos, Portugal; 4School of Resources and Environment, Henan Institute of Science and Technology, 90 East of Hualan Avenue, Xinxiang 453003, China

**Keywords:** insect pest control, fall armyworm, chemical pesticides, insecticidal dsRNA, *V-ATPaseD*

## Abstract

*Spodoptera frugiperda* is a major global pest that significantly impacts key crops. While chemical pesticides are the primary control method, their prolonged use fosters resistance and environmental harm. RNA interference (RNAi) offers a promising alternative but is limited by the instability of double-stranded RNA (dsRNA) and its delayed action. Chitosan-based delivery systems improve dsRNA stability, and combining RNAi with pesticides may enhance efficacy. Here, the vacuolar-type H+-ATPase (V-ATPase), an essential multi-subunit enzyme complex critical for insect development and nutrient uptake, was selected as the potential RNAi target, and *V-ATPaseD* silencing was achieved through topical or oral administration of chitosan/dsRNA-V-ATPaseD nanoparticles. Combining chitosan/dsRNA-V-ATPaseD nanoparticles with emamectin benzoate–lufenuron significantly enhanced pest control efficacy, highlighting the potential of integrating RNAi into conventional pesticides to create sustainable pest management strategies.

## 1. Introduction

The fall armyworm, *Spodoptera frugiperda* (J.E. Smith), a lepidopteran pest from the family Noctuidae, is native to tropical and subtropical regions of the Americas [1]. This highly polyphagous insect feeds on over 350 plant species, including staple crops like maize and rice, causing significant agricultural losses [2]. Since 2016, *S. frugiperda* has expanded its range to Africa, Asia, and Oceania [3], where it has become a major invasive pest.

Chemical insecticides, including lufenuron, chlorantraniliprole, and abamectin, remain the primary tools for managing *S. frugiperda* [4]. Beyond chemical control, biological control agents such as parasitoids and predators [5,6] and genetically engineered Bt crops expressing *Bacillus thuringiensis* (Bt) toxins [7,8], have also been widely used to combat this pest. More recently, RNA interference (RNAi) technology has emerged as a promising alternative, offering a novel and highly specific mechanism for pest control by silencing essential genes using target-specific double-stranded RNA (dsRNA) molecules [9,10]. Notably, RNAi pesticides, now classified under IRAC MoA group 35 (RNAi-mediated target suppressors), represent a significant advancement in pest management, as demonstrated by the commercial success of GreenLight’s Calantha, which targets the Colorado potato beetle [11].

RNAi pesticides offer several advantages over conventional pesticides, including greater specificity, reduced resistance risk, lower environmental toxicity, and decreased development costs [4,12]. However, their adoption faces challenges, particularly the instability of dsRNA and difficulties with effective delivery. To address these limitations, nanocarriers such as chitosan have been developed to stabilize dsRNA and enhance the efficiency of its delivery [13,14]. These systems hold promise for reducing pesticide use while ensuring effective pest control. For example, CYR-Chitosan/siRNA nanoparticles, which encapsulate cyromazine (CYR) pesticides and short-interference RNA (siRNA), targeting the *S. frugiperda* chitin synthase (*SfCHS*) gene involved in cuticle formation, achieved a high mortality with minimal pesticide use [14]. Similarly, in the rice pest *Nilaparvata lugens*, the chitosan-modified nanomaterial ROPE@C successfully delivered dsRNA targeting the chitin synthetase A (*NlCHSA*) gene, significantly reducing survival rates [15].

Among potential RNAi targets, vacuolar-type H+-ATPase (V-ATPase) stands out as an essential multi-subunit enzyme complex critical for insect development and nutrient uptake [16]. The V1 domain of V-ATPase, comprising subunits A-H, drives proton translocation via ATP hydrolysis, facilitating epithelial transport and nutrient absorption in the insect midgut [17,18]. Previous studies targeting V-ATPaseA and B subunits in *Spodoptera* spp. and other pests have demonstrated promising outcomes, including reduced gene expression and increased larval mortality [19,20]. However, less attention has been paid to other subunits of the proton transport pump, which are critical for the enzyme’s structure and function [21].

Conventional pesticides such as lufenuron, chlorantraniliprole, and abamectin, though effective, are increasingly challenged by the development of resistance and pose risks to non-target organisms [22,23,24]. Combining RNAi technology with chemical pesticides offers a promising approach to enhance pest control efficacy while reducing chemical inputs, potentially addressing these limitations.

In this study, we use RNAi to target the *V-ATPaseD* in *S. frugiperda* and evaluate the additive effects of chitosan/dsRNA-V-ATPaseD nanoparticles and an emamectin benzoate–lufenuron formulation in bioassays, with the dsRNA treatment applied first, followed by the chemical insecticide after 24 h. To control for off-target effects, dsRNA targeting the enhanced green fluorescent protein (*eGFP*) gene was used as a non-target control in all RNAi experiments. Our results provide new insights into pesticide reduction and sustainable pest control strategies for this invasive species.

## 2. Materials and Methods

### 2.1. Insect Rearing and Maintenance

The strain of *S. frugiperda* used was obtained from Keyun Bio. Ltd. (Jiaozuo China) and has been maintained in our laboratory for over ten generations. Insects at all developmental stages were reared under controlled conditions of 28 ± 1 °C, 60% ± 10% relative humidity, and a 14:10 h (L:D) photoperiod. Larvae were fed an artificial diet specific to *S. frugiperda*, purchased from Keyun Bio. Ltd. (Jiaozuo, China) (https://3.cn/-2dJbK2H, accessed on 24 December 2022). After reaching the third instar, larvae were individually placed in labeled containers. Adults were provided with a 10% honey solution.

### 2.2. Insecticide and Reagents

Emamectin benzoate–lufenuron (0.1 g/mL) was kindly supplied by Xifunong Biotechnology Co. (Kaifeng, China). Acetone was purchased from Amresco Co. (Framingham, MA USA). Chitosan and sodium sulfate were acquired from Sigma-Aldrich (St. Louis, MO, USA). Sodium hydroxide and glacial acetic acid were obtained from Sinopharm Chemical Reagent Co., Ltd. (Shanghai, China).

### 2.3. Total RNA Isolation and cDNA Synthesis

Total RNA was isolated from the entire larval body (with intestinal contents removed) using the TRIzol method (Thermo Fisher, Waltham, MA, USA). Prior to RNA extraction, dissected larvae were flash frozen in liquid nitrogen and then homogenized using a mortar and pestle. To minimize individual variability and ensure representative results, RNA samples were pooled from three larvae per replicate for all qRT-PCR experiments. RNA quality and concentration were assessed using a NanoDrop spectrophotometer (Thermo Fisher, Waltham, MA, USA) and agarose gel electrophoresis. First-strand cDNA was synthesized from the purified RNA using the PrimeScript™ II First Strand cDNA Synthesis Kit (TaKaRa, Dalian, Liaoning, China), following the manufacturer’s instructions.

### 2.4. Quantitative Real-Time PCR Analysis

Quantitative real-time PCR (qRT-PCR) was conducted following the protocol outlined by Wang et al. (2023) [25]. qRT-PCR reactions (20 μL) were carried out with the CFX96™ Real-Time PCR Detection System (Bio-Rad, Hercules, CA, USA) using the FS Universal SYBR Green Master Mix (Roche, Cornwall, England, UK) according to the manufacturer’s instructions. Each reaction consisted of 10 μL of the 2× concentrated FS Universal SYBR Green Master Mix, 300 nM of each primer, and 50 ng of template cDNA prepared as described in Section 2.3. Transcript levels of *V-ATPaseD* were normalized using a *β-actin* amplicon (GenBank Acc. no. MN044625.1) from *S. frugiperda* [26]. Data analysis was performed using the 2^−△Ct△Ct^ method [27]. qRT-PCR primers for *V-ATPaseD* and *β-actin* are listed in Table 1.

### 2.5. Synthesis of dsRNA

The DNA template for the dsRNA synthesis of the *V-ATPaseD* gene (GenBank Acc. No. MT707617.1) was amplified by PCR using cDNA synthesized from the total RNA of third instar *S. frugiperda* larvae, prepared as described in Section 2.3. Similarly, the DNA template for the dsRNA synthesis of the *eGFP* gene (GenBank Acc. No. MH070103.1), used as an off-target control, was amplified from the piggyBac[*eGFP*] vector, previously constructed in our laboratory. Gene-specific primers were designed and extended with T7 promoter sequences (Table 1). The dsRNA for *V-ATPaseD* (dsRNA-V-ATPaseD) and *eGFP* (dsRNA-eGFP) was synthesized in vitro using the MEGAscript RNAi Kit (Thermo Fisher, Waltham, MA, USA), following the manufacturer’s instructions.

### 2.6. RNAi by Injection

The dsRNA-V-ATPaseD and the nonspecific control dsRNA-eGFP were injected into the hemolymph of 2-day-old fourth-instar larvae, with each larva receiving a dose of 5 μg. Three independent experiments were conducted, each consisting of 10 larvae per treatment group. Larval phenotypic changes, mortality rates, and weights were monitored every 24 h post-injection until pupation (24 h, 48 h, 72 h, 96 h, 120 h, 144 h, and 168 h). For gene expression analysis, total larval RNA extraction and cDNA synthesis were conducted as described in Section 2.3. The resulting cDNA was then used for qRT-PCR following the protocol outlined in Section 2.4. Three independent experiments were conducted, each with ten larvae per treatment group.

### 2.7. Preparation of Chitosan/dsRNA Nanocomplexes

The chitosan/dsRNA nanocomplexes were prepared following the method described by Sandal et al. [28], with minor modifications. In brief, 0.2 g of chitosan was dissolved in 100 mL of 0.1 M NaAc buffer (0.1 M NaC_2_H_3_O_2_–0.1 M acetic acid, pH 4.5) to prepare a 0.2% chitosan solution. Next, 100 μL of the 0.2% chitosan solution was mixed with 900 μL of 0.1 M Na_2_SO_4_ buffer and then combined with 200 μg of dsRNA-V-ATPaseD or dsRNA-eGFP dissolved in 1 mL DEPC-treated water. The mixture was incubated at 55 °C for 1 min, vortexed for 30 s, and subsequently incubated at room temperature for 1 h to allow nanocomplex formation. The resulting nanocomplexes were centrifuged at 12,000 rpm for 10 min, and the pellets were resuspended in 40 μL of DEPC-treated water.

### 2.8. RNAi by Feeding

Chitosan/dsRNA-V-ATPaseD and Chitosan/dsRNA-eGFP nanocomplexes were applied to the surface of the artificial diet at a concentration of 5 μg per 9 mm^2^. Phenotypic changes, mortality rates, and larval weights were recorded at 24 h intervals post-exposure (24 h, 48 h, 72 h, 96 h, 120 h, 144 h, 168 h and 192 h). For gene expression analysis, larval total RNA extraction and cDNA synthesis were conducted as described in Section 2.3. The resulting cDNA was then used for qRT-PCR following the protocol outlined in Section 2.4. Three independent experiments were conducted, each with ten larvae per treatment group.

### 2.9. RNAi by Topical Delivery

Chitosan/dsRNA-V-ATPaseD and chitosan/dsRNA-eGFP nanocomplexes were topically applied to the dorsal surface (tergum) of the fourth ventral segment of larvae at a dose of 5 μg per larva. After the droplet was fully absorbed, the larvae were returned to the artificial diet. Phenotypic changes, mortality rates, and larval weights were monitored and recorded at 24 h intervals post-application until pupation (24 h, 48 h, 72 h, 96 h, 120 h, 144 h, 168 h, and 192 h). For gene expression analysis, total larval RNA extraction and cDNA synthesis were conducted as described in Section 2.3. The resulting cDNA was then used for qRT-PCR following the protocol outlined in Section 2.4. Three independent experiments were conducted, each with ten larvae per treatment group.

### 2.10. Bioassays with Chemical Insecticides

Emamectin benzoate–lufenuron was prepared in 20% acetone at concentrations of 25, 50, 75, and 100 mg/L. These solutions were uniformly sprayed onto Petri dishes containing both artificial diet and larvae. Phenotypic changes, mortality rates, and larval weights were recorded every 24 h until pupation (24 h, 48 h, 72 h, and 96 h). Mortality data were used to construct Kaplan–Meier survival curves, and mortality rates at 48 h post-treatment were analyzed to determine the median lethal concentration (LC_50_) and sublethal concentration (LC_30_). Each treatment was performed in triplicate, with 10 larvae per replicate.

### 2.11. Combined Exposure to dsRNA Nanocomplex and Chemical Pesticide

Fourth-instar larvae were initially fed chitosan/dsRNA-V-ATPaseD or chitosan/dsRNA-eGFP nanocomplexes. Twenty-four hours after dsRNA treatment, an emamectin benzoate–lufenuron formulation, prepared at its LC_30_ concentration, was uniformly applied to Petri dishes containing the artificial diet and larvae. Phenotypic changes, mortality rates, and larval weights were monitored and recorded at 24 h intervals following the pesticide application (24 h, 48 h, 72 h, 96 h, 120 h, and 144 h). Mortality data were used to construct Kaplan–Meier survival curves. Each treatment was conducted in triplicate, with 10 larvae per replicate.

### 2.12. Data Analysis

Data analysis was conducted using SPSS Statistics 26.0 for statistical tests and GraphPad Prism 8.0.2 for data visualization. Statistical significance (*p* < 0.05) in qRT-PCR and larval weight measurements was assessed using Student’s *t*-tests for parametric data or Mann–Whitney U tests for nonparametric data, based on the Shapiro–Wilk test for normality. For multiple comparison analyses, the Bonferroni correction was applied to control for Type I errors.

The survival rate of *S. frugiperda* was calculated as the percentage of individuals surviving over a specific period after treatment. Survival data were analyzed using the Kaplan–Meier method, with differences between survival curves assessed using the log-rank (Mantel–Cox) test. Statistical significance was defined as *p* < 0.05.

Based on our preliminary results, pesticide concentrations for bioassays were selected to achieve mortality rates of 20% to 90% after 48 h of treatment. The LC_50_ and LC_30_ values, along with their 95% confidence intervals (CIs), were determined through logistic regression of the mortality data using R software (version 4.3.3) [29]. Mortality proportions were calculated as the ratio of dead larvae to the total number of larvae at each pesticide concentration. A generalized linear model (GLM) with a binomial error distribution was fitted, using log-transformed pesticide concentrations as predictors. Overdispersion was accounted for with a quasibinomial adjustment. The LC_50_ and LC_30_ values were estimated using the dose.p function in the MASS package (version 7.3-60 2023) [30], with confidence intervals derived from the model. Visualization of the fitted model was performed using R’s base plotting tools for clarity and precision. Each treatment was conducted in triplicate, with 10 larvae per replicate.

## 3. Results

### 3.1. RNAi of V-ATPaseD via dsRNA Injection

The structural model of the V-ATPase enzyme highlights its two main domains: the V1 complex, responsible for ATP hydrolysis, and the V0 complex, which facilitates proton translocation (Figure 1A, www.doccheck.com accessed on 1 January 2018.). Within the V1 complex, subunit D, in conjunction with subunits C and H, forms a structural bridge connecting the V1 and V0 domains. This connection is crucial for maintaining cellular functions such as ion transport and pH homeostasis. The pivotal role of V-ATPaseD in maintaining the structural and functional integrity of the V-ATPase complex suggests that RNAi-mediated gene silencing of this subunit could effectively disrupt V-ATPase activity, positioning it as a promising candidate for RNAi-based control strategies. Expression analysis of *V-ATPaseD* in *S. frugiperda* larvae revealed its presence across all six larval instars, with instar-specific fluctuations and distinct expression peaks (Figure 1B).

To evaluate the functional impact of *V-ATPaseD* knockdown, RNAi experiments were conducted by injecting the specific dsRNA-V-ATPaseD into the hemolymph of 2-day-old fourth-instar larvae, with nonspecific dsRNA-eGFP serving as an off-target control. The RNAi treatment significantly reduced *V-ATPaseD* mRNA levels at 24 and 48 h post-injection (Figure 1C), confirming the efficacy of the RNAi response. Knockdown of *V-ATPaseD* led to developmental abnormalities in some treated larvae, including molting defects and impaired pupation observed in 23.33% of treated individuals (Figure 1D). In contrast, larvae treated with dsRNA-eGFP showed no significant effects.

### 3.2. RNAi of V-ATPaseD via Topical Administration of Chitosan/dsRNA Nanocomplexes

Chitosan/dsRNA-V-ATPaseD and chitosan/dsRNA-eGFP nanocomplexes were applied to the tergum of 2-day-old fourth-instar larvae to evaluate the efficacy of topical RNAi delivery. The relative *V-ATPaseD* mRNA levels in treated and control larvae were monitored over 120 h. No significant difference in *V-ATPaseD* expression was observed between treated and control groups at 24 and 48 h, indicating the absence of knockdown during these early time points. However, by 72 h post-treatment, a significant reduction in *V-ATPaseD* expression (*p* < 0.05) was detected in the treated group, suggesting a delayed onset of RNAi-mediated knockdown (Figure 2A). This suppression persisted until 96 h, with a statistically significant decrease in mRNA levels (*p* < 0.05). By 120 h, *V-ATPaseD* expression in treated larvae returned to levels comparable to the controls, indicating partial recovery of transcript levels. Notably, larvae treated with chitosan/dsRNA-V-ATPaseD exhibited significant weight loss compared to those treated with chitosan/dsRNA-eGFP at 120 h (Figure 2B). Additionally, two out of thirty larvae died during pupation, resulting in a mortality rate of 6.67%. In contrast, no mortality was observed in the control chitosan/dsRNA-eGFP group (Figure 2C).

### 3.3. RNAi of V-ATPaseD via Feeding Chitosan/dsRNA Nanocomplexes

To assess the efficacy of RNAi via oral delivery, *S. frugiperda* larvae were fed chitosan/dsRNA nanocomplexes targeting V-ATPaseD or the off-target control eGFP. Feeding on chitosan/dsRNA-V-ATPaseD resulted in a transient, time-dependent reduction in V-ATPaseD mRNA levels (Figure 3A). A significant knockdown was observed at 72 h post-feeding (*p* < 0.001) followed by a recovery to baseline levels by 96 h.

Although no significant differences in larval weight were observed between treatments from 72 h onward (Figure 3B), larvae exposed to chitosan/dsRNA-V-ATPaseD also displayed impaired pupation, with mortality rate of 20% at 192 h compared to the control group fed chitosan/dsRNA-eGFP (3.33%). These abnormalities further underscore the critical role of *V-ATPaseD* in the growth and development of *S. frugiperda* (Figure 3C).

### 3.4. Mortality and Growth Inhibition by Emamectin Benzoate–Lufenuron on S. frugiperda

The efficacy of the emamectin benzoate–lufenuron pesticide formulation in inducing mortality and growth inhibition was assessed by exposing 2-day-old, fourth-instar *S. frugiperda* larvae to concentrations of 25, 50, 75, and 100 mg/L. Larval weight measurements taken every 24 h indicated a significant weight reduction in treated larvae compared to controls, especially at higher concentrations (Figure 4A). Treated larvae also resulted in notable disruptions to larval development (Figure 4B). Additionally, phenotypic observations revealed reduced activity and feeding disruption due to pesticide exposure. Mortality rates increased with pesticide concentration, and survival curves showed statistically significant differences (Mantel–Cox test, *p* < 0.001) among the pesticide concentrations (Figure 4C).

Bioassay results at 48 h confirmed a dose-dependent increase in the mortality of *S. frugiperda* larvae exposed to emamectin benzoate–lufenuron, as indicated by logistic regression analysis (Figure 4D). Mortality ranged from less than 0.2 at 25 mg/L to nearly 0.8 at 100 mg/L. The control group (non-sprayed) showed negligible mortality, demonstrating the treatment’s effectiveness. The LC_50_ was estimated at 53.03 µg/mL (log-transformed: 3.97), with a 95% confidence interval of [43.27, 64.98], while the sublethal concentration LC_30_ was determined to be 34.75 µg/mL (log-transformed: 3.55), with a 95% confidence interval of [28.36, 42.58]. The fitted model equation was logit(*p*) = −7.96 + 2 × log(conc). Both the intercept (−7.96) and slope (2) coefficients were highly significant (*p* < 0.001), highlighting their substantial contribution to the model and the strong impact of concentration on mortality. The logistic regression model demonstrated a good fit, reflected by the considerable reduction in deviance from null deviance (28.4551) to residual deviance (1.9937). An AIC value of 20.354 further supports the model’s quality, indicating a balanced trade-off between fit and complexity. The model converged efficiently within four Fisher scoring iterations.

### 3.5. Additive Effects of Chitosan/dsRNA-V-ATPaseD Nanocomplex and Emamectin Benzoate-Lufenuron Against S. frugiperda

To evaluate the synergistic interaction between chitosan/dsRNA-V-ATPaseD nanoparticles and the chemical pesticide, the nanoparticles were incorporated into the artificial diet and provided to the 2-day-old fourth-instar *S. frugiperda* larvae. After 24 h of exposure, larvae were sprayed with emamectin benzoate–lufenuron at its LC_30_ concentration (34.75 µg/mL). The survival curves illustrate the effects of the various treatments on *S. frugiperda* larvae (Figure 5). Larvae in the control (unexposed) group (CK) exhibited no significant mortality over the observation period, maintaining a nearly 100% survival rate. Exposure to the sublethal concentration (LC_30_) of the chemical pesticide alone resulted in a gradual decrease in survival, comparable to larvae treated with LC_30_ combined with off-target RNAi control (*eGFP*). In contrast, larvae exposed to the combined treatment of LC_30_ and dsRNA-V-ATPaseD exhibited a significantly accelerated decline in survival at 48 (Mantel–Cox test, *p* < 0.05) and 72 (Mantel–Cox test, *p* < 0.05) hours. At these time points, the survival rate of the LC_30_ + dsRNA-V-ATPaseD group was significantly lower than both the LC_30_-alone and LC_30_ + eGFP groups (Mantel–Cox test, *p* < 0.05), resulting in 68% mortality—27% higher than the pesticide alone—72 h post-exposure. This indicates faster mortality and additive effects between the RNAi-mediated gene silencing and the pesticide. Overall, the enhanced efficacy of the combined treatment accelerated insect mortality within the first 72 h through a mechanism that remains to be fully elucidated.

## 4. Discussion

In the present study, we investigated the RNA interference (RNAi) of *S. frugiperda* V-ATPase subunit D focusing on its potential as a novel target for RNAi-based pesticides. This subunit is crucial for linking the V1 and V0 domains of the proton pump, yet it has been largely overlooked in prior RNAi research. Specifically, subunit D is a component of the rotational (rotor) subcomplex, contributing to the stabilization of the entire V-ATPase structure [31,32]. Our findings identified *V-ATPaseD* as a particularly promising target, showing the highest level of mRNA suppression at 24 h post-injection of specific dsRNA. Also, RNAi of *V-ATPaseD* produced molting defects and reduced pupation rates, indicating the V-ATPase’s essential role in maintaining physiological processes for insect development. In a previous study, RNAi targeting *V-ATPaseD* in *Liriomyza trifolii* larvae similarly showed peak suppression at 24 h post-injection of specific dsRNA, with recovery to baseline levels by 48 h and only 2% of treated adults surviving after five days [19].

Our study demonstrated that chitosan/dsRNA-V-ATPaseD nanoparticles, administered both topically and via feeding, successfully downregulated *V-ATPaseD* expression and impaired larval fitness, by reducing pupation rates. RNAi effects were first detected 72 h post-treatment with both administration routes. Notably, gene silencing took longer using these routes compared to direct injection. This delay may be associated with the amount of dsRNA ingested and the biological barriers it must overcome, such as the alkaline pH of the midgut and the peritrophic matrix [33]. Consequently, the effectiveness of oral RNAi depends on both larval feeding behavior and dsRNA stability in the digestive system. Chitosan’s ability to stabilize dsRNA and protect it from gut nucleases preserves the RNAi effect over time, enabling sustained gene suppression and enhancing long-term pest control potential [34].

However, the efficacy of chitosan nanoparticles in insects like *S. frugiperda* can be influenced by biological and physiological barriers. The alkaline pH of the lepidopteran midgut, for instance, may affect the stability of chitosan nanoparticles, potentially limiting their effectiveness [35]. Furthermore, the presence of the peritrophic matrix, a protective layer in the insect gut, poses an additional challenge for dsRNA delivery. To overcome these obstacles, further optimization of nanoparticle size and formulation (e.g., chitosan: dsRNA ratio) is necessary [36]. Additionally, chemical modifications to chitosan, such as PEGylation, or surface modifications, could enhance the stability of the nanoparticles in the alkaline gut environment and potentially enhance their ability to cross biological barriers and improve dsRNA uptake [37].

In our study, the co-application of chitosan/dsRNA-V-ATPaseD with the chemical insecticide emamectin benzoate–lufenuron resulted in an additive effect, significantly increasing the mortality of *S. frugiperda* larvae. This combined approach offers several advantages. While farmers traditionally favor chemical pesticides for their rapid action, the growing issue of pesticide resistance is leading to an increase in pesticide doses, posing risks to human health and the environment. By integrating RNAi-based methods with conventional insecticides, as demonstrated here, it is possible to maintain rapid pest control while reducing chemical inputs. This strategy not only enhances pest suppression but also contributes to the management of pesticide resistance by reducing the need for higher doses of chemical agents.

Looking forward, the combined use of RNAi pesticides and conventional insecticides represents a promising strategy for sustainable pest management. Innovations in nanoparticle design and RNAi delivery systems will further improve the stability and efficacy of these approaches, enabling a gradual reduction in the use of conventional pesticides. Over time, this could lead to more environmentally friendly pest control solutions that minimize chemical exposure to humans and reduce the ecological impact on agricultural systems.

## 5. Conclusions

This study demonstrated the successful knockdown of the V-ATPase subunit D in *S. frugiperda* larvae through direct dsRNA injection and topical or oral delivery of chitosan/dsRNA nanoparticles. Suppression of the expression of *V-ATPaseD* resulted in reduced pupation rates, highlighting its potential as a target for RNAi-based pest control in this invasive species. Notably, combining chitosan/dsRNA-V-ATPaseD nanoparticles with sublethal concentrations of the insecticide emamectin benzoate–lufenuron produced an additive effect, significantly increasing larval mortality compared to insecticide treatment alone. These findings demonstrate the promise of integrating RNAi nanocarriers with conventional insecticides to enhance pest control efficacy, reduce pesticide reliance, and address challenges such as resistance management and environmental sustainability.

## Figures and Tables

**Figure 1 insects-16-00348-f001:**
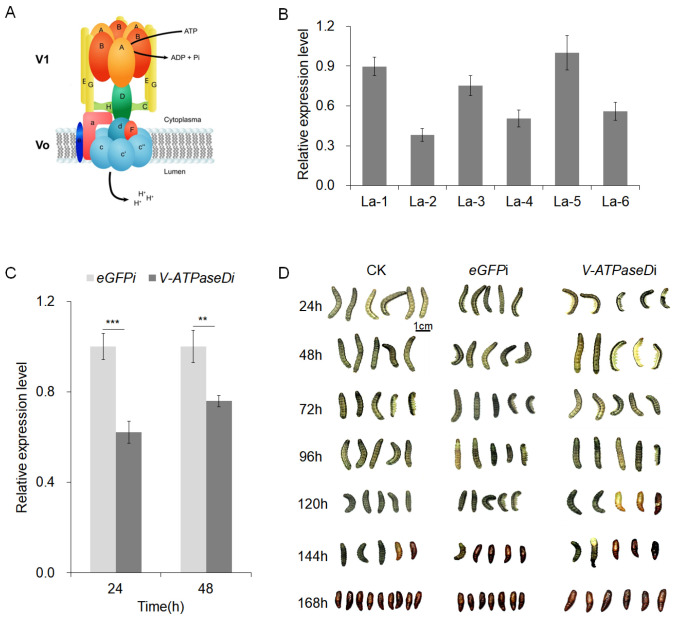
Characterization and expression of *V-ATPaseD* as well as its RNAi-mediated knockdown effects in *S. frugiperda*. (**A**) Schematic representation of the V-ATPase complex, illustrating the V1 domain responsible for ATP hydrolysis and the V0 domain for proton translocation across the membrane. The V1 domain of V-ATPase, comprising subunits A, B, C, D, E, G and H, the V0 domain of V-ATPase, comprising subunits a, c, c′, c″, d, e and F. Subunit D, a critical component of the V1 domain, is highlighted in green. (**B**) Expression profile of the *V-ATPaseD* gene across the six larval instars (La-1 to La-6), analyzed by qRT-PCR. *β-Actin* was used as the internal reference gene, and the La-5 set as the reference condition (relative expression = 1.0). (**C**) RNAi-mediated knockdown of *V-ATPaseD* mRNA following dsRNA-V-ATPaseD injection compared to dsRNA-eGFP injection (off-target control). Expression levels were assessed at 24 and 48 h post-treatment by qRT-PCR. A significant reduction is observed at 24 h (*** *p* < 0.001) and 48 h (** *p* < 0.01). Data are presented as mean ± SEM from three independent experiments, each conducted in triplicate. (**D**) Representative images showing the developmental progression of *S. frugiperda* larvae following RNAi treatment with dsRNA-eGFP (*eGFPi*) or dsRNA-V-ATPaseD (*V-Apisai*). Larvae were photographed at 24 h intervals from 24 to 168 h post-injection, revealing molting defects in the V-ATPaseD-treated group compared to the control. CK represents the untreated control group for larval experiments. The scale bar represents 1 cm.

**Figure 2 insects-16-00348-f002:**
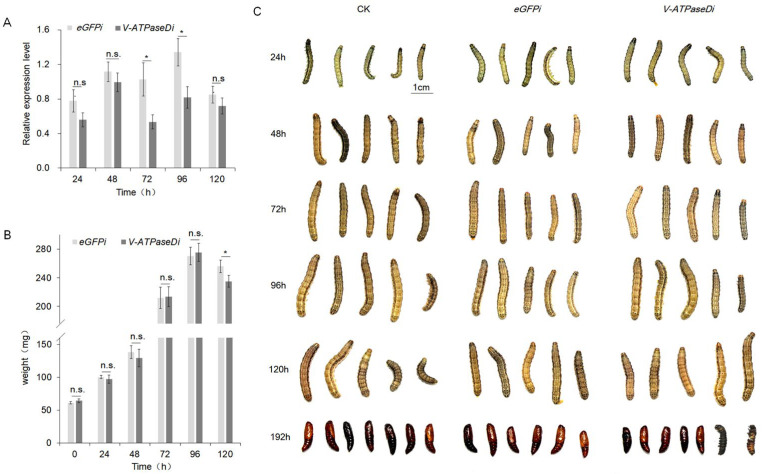
RNAi of *V-ATPaseD* via topical application of chitosan/dsRNA nanocomplex. (**A**) qRT-PCR analysis of *V-ATPaseD* transcript levels in larvae treated with dsRNA-V-ATPaseD compared to those treated with dsRNA-eGFP (off-target control). Samples for qRT-PCR were collected at 24, 48, 72, 96, and 120 h post-treatment. Expression levels were normalized using β-actin as the reference gene. Data represent means ± SEM from three independent experiments, each performed in triplicate. Significance analysis was conducted with Student’s *t*-test (* *p* < 0.05, n.s. *p* > 0.05). (**B**) Mean larval weights were recorded at the same time points following treatment with dsRNA-V-ATPaseD or dsRNA-eGFP. A significant reduction in larval weight was observed at 120 h post-treatment (* *p* < 0.05, n.s. *p* > 0.05). (**C**) Representative images illustrating the morphological and developmental stages of *S. frugiperda* larvae and pupae of insects treated with dsRNA-V-ATPaseD, dsRNA-eGFP, and the untreated control (CK). Images were captured at 24, 48, 72, 96, 120, and 192 h post-treatment. The scale bar represents 1 cm.

**Figure 3 insects-16-00348-f003:**
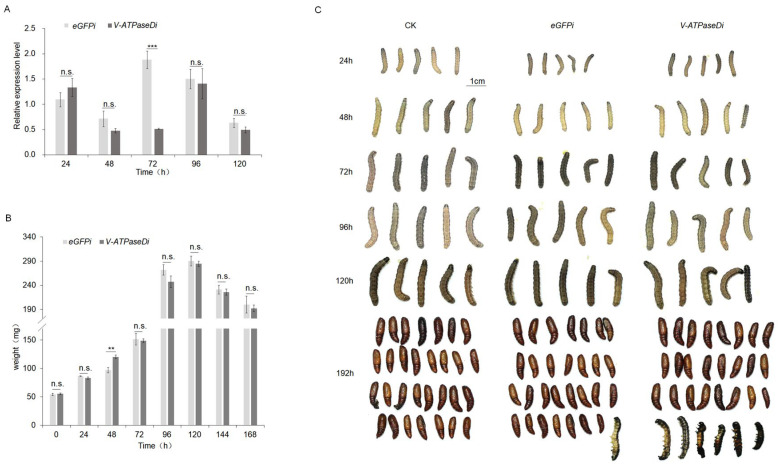
RNAi of *V-ATPaseD* via oral administration of chitosan/dsRNA nanocomplex. (**A**) qRT-PCR analysis of *V-ATPaseD* transcript levels in larvae following oral administration of dsRNA-V-ATPaseD, compared to the off-target control (dsRNA-eGFP). Samples for qRT-PCR were collected at 24, 48, 72, 96, and 120 h post-treatment. Expression levels were normalized to *β-Actin* as the internal reference gene. Statistically significant differences are denoted by asterisks (*** *p* < 0.001), and non-significant differences are marked as n.s. (**B**) Larval weights were recorded at 24 h intervals post-treatment. A significant reduction in weight was observed at 48 h (** *p* < 0.01) in larvae treated with dsRNA-V-ATPaseD compared to dsRNA-eGFP, while no significant differences were observed at other time points (n.s.). (**C**) Morphological and developmental effects: Representative images of larvae and pupae from untreated controls (CK), dsRNA-eGFP-treated, and dsRNA-V-ATPaseD-treated groups at 24, 48, 72, 96, 120, and 192 h post-treatment. Larvae treated with dsRNA-V-ATPaseD exhibited a significant mortality rate 20% at 192 h compared to the control group fed chitosan/dsRNA-eGFP (3.33%). The total number of pupae obtained for each treatment is shown in the image, which reflects the cumulative results from all replicates. Scale bar = 1 cm.

**Figure 4 insects-16-00348-f004:**
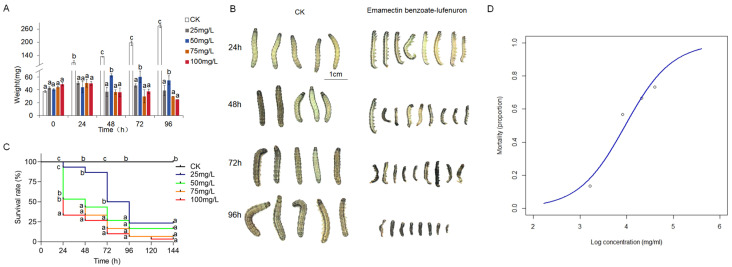
Toxicity of emamectin benzoate–lufenuron in *S. frugiperda* administered via spray. (**A**) Weight measurements of 2-day-old, 4th-instar *S. frugiperda* larvae treated with different concentrations of emamectin benzoate–lufenuron (25, 50, 75, and 100 mg/L) compared to the untreated control (CK) at 24, 48, 72, and 96 h post-treatment. A significant reduction in weight was observed in a dose- and time-dependent manner, as indicated by alphabetical letters (a, b, c) from multiple comparison analyses. (**B**) Representative images of surviving *S. frugiperda* larvae at 24, 48, 72, and 96 h post-treatment with emamectin benzoate–lufenuron compared to CK. Larvae treated with the insecticide exhibited visible growth inhibition and morphological abnormalities. Scale bar = 1 cm. (**C**) Kaplan–Meier survival curves showing the dose-dependent reduction in survival rate of larvae treated with emamectin benzoate-lufenuron at various concentrations over 7 days. Each experiment was replicated three times with ten individuals per replicate. Data represent means ± SEM from three independent experiments. Differences between survival curves were analyzed using the log-rank (Mantel–Cox) test. Different letters denote statistically significant differences (*p* < 0.05). (**D**) Mortality (proportion) of *S. frugiperda* larvae as a function of the log-transformed emamectin benzoate–lufenuron concentration (µg/mL). The blue line represents the predicted values from the minimal adequate statistical model fitted to the emamectin benzoate-lufenuron bioassay, demonstrating a significant dose-dependent response.

**Figure 5 insects-16-00348-f005:**
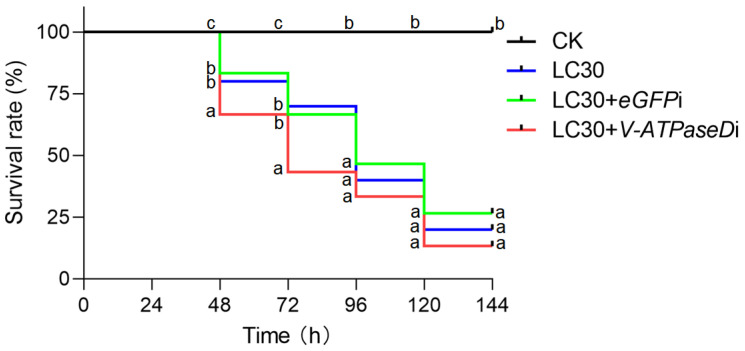
Additive effect of chitosan/dsRNA-V-ATPaseD nanoparticles and sublethal (LC_30_) emamectin benzoate–lufenuron on *S. frugiperda*. Kaplan–Meier survival curves illustrating the survival rates of 2-day-old 4th instar *S. frugiperda* larvae treated with sublethal concentrations (LC_30_) of emamectin benzoate–lufenuron alone (blue line), in combination with control dsRNA (eGFP, green line), and in combination with dsRNA targeting V-ATPaseD (red line). The combined treatment of LC_30_ and dsRNA-V-ATPaseD significantly reduced larval survival compared to other treatments, demonstrating an additive effect. Untreated larvae (CK, black line) served as the control. Differences between survival curves were analyzed using the log-rank (Mantel–Cox) test. Different letters denote statistically significant differences (*p* < 0.05).

**Table 1 insects-16-00348-t001:** Primer sets designed and used in this study. The underlined sequences in the primers used to generate DNA templates for the in vitro synthesis of dsRNA (as described in Section 2.5) correspond to the 24-nucleotide T7 RNA polymerase promoter added at the 5′ end.

Gene	Forward Primer (5′-3′)	Reverse Primer (5′-3′)	Utility
*V-ATPaseD*-dsRNA (MT707617.1)	**GAAATTAATACGACTCACTATAGG**CCTCCAAGTGAGGTTCCGTA	**GAAATTAATACGACTCACTATAGG**CAACTCGACCAGCAGTTTCA	Primers for synthesis of *V-ATPaseD* dsRNA
*eGFP*-dsRNA (MH070103.1)	**GAAATTAATACGACTCACTATAGG**GTACGGCGTGCAGTGCT	**GAAATTAATACGACTCACTATAGG**GTGATCGCGCTTCTCG	Primers for synthesis of *eGFP* dsRNA
*V-ATPaseD* (MT707617.1)	TCGCTTACATCATCTCCG	AACAGCAGGTCCTCGTCA	qRT-PCR primers for *V-ATPaseD*
*β-actin* (MN044625.1)	GATGTCGGGACGGGATA	TCATACGGCGAGTGCTT	qRT-PCR primers for *β-actin*

## Data Availability

The original contributions presented in this study are included in the article. Further inquiries can be directed to the corresponding authors.

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
