# Peer review of "Additive Insecticidal Effects of Chitosan/dsRNA Nanoparticles Targeting V-ATPaseD and Emamectin Benzoate–Lufenuron Formulations Against Spodoptera frugiperda (J.E. Smith) (Lepidoptera: Noctuidae)"

_insects, 2025, doi:10.3390/insects16040348_

Round 1
Reviewer 1 Report (Previous Reviewer 1)
Comments and Suggestions for Authors
The authors have addressed all the comments by the reviewers and made appropriate improvements were necessary. Some additional information and discussion, described below, would further improve the manuscript and make the methods and conclusions clearer.
Comment regarding diet composition is not fully resolved. Keyun Bio Ltd webpage does not list S. frugiperda diet as a product for purchase. To make methods reproducible, composition of diet must be explicit, a reference another article with the explicit composition included, or a link provided where the specific diet could be purchased.
Authors responded to comment regarding tissue used for RNA extraction, but it highlights another issue. Expression of the target genes may be quite different between tissues, and the RNAi effect may not be systemic for some delivery methods, particularly oral dsRNA, due to the barriers identified and mentioned by the authors in the discussion. Results of whole body TRIzol extraction may be masking actual results of silencing efficiency. This should be an important point to add to the discussion of the RT-qPCR results. Also, justification of why whole larvae were used instead of specific tissues (such as midgut for oral delivery) is important, since it introduces complications for data analysis and interpretation. Fourth instar S. frugiperda larvae are sufficiently large to do tissue specific dissections with ease. Additional question: were larvae directly homogenized/processed in Trizol, what equipment was used?
Text description of figure 1C still indicates that no significant difference was found at 48 hrs, despite the graph indicating otherwise. 1D: if representative images are were used according to author response, indicate this.
Figures 1, 2 and 3. How many larvae were used for TRIzol extraction for each sample? If it is a pool of several larvae, differences in efficiency between larvae may also contribute to masking RT-qPCR results. Not all larvae are silenced equally.
Author Response
C1: Comment regarding diet composition is not fully resolved. Keyun Bio Ltd webpage does not list S. frugiperda diet as a product for purchase. To make methods reproducible, composition of diet must be explicit, a reference another article with the explicit composition included, or a link provided where the specific diet could be purchased.
Response: Thank you for your comment. We acknowledge the importance of providing detailed and reproducible methods, including the composition of the diet. However, the company, Keyun Bio Ltd, does not disclose the specific composition of the S. frugiperda diet as it is considered proprietary information. To address this concern, we have included the direct link to the product page where the diet can be purchased (https://3.cn/-2dJbK2H) in Line 126, Page 3 of revised manuscript. This ensures that other researchers can access the same diet used in our experiments. We hope this clarification and the provided link sufficiently address the reviewer’s concern regarding reproducibility.
C2: Authors responded to comment regarding tissue used for RNA extraction, but it highlights another issue. Expression of the target genes may be quite different between tissues, and the RNAi effect may not be systemic for some delivery methods, particularly oral dsRNA, due to the barriers identified and mentioned by the authors in the discussion. Results of whole body TRIzol extraction may be masking actual results of silencing efficiency. This should be an important point to add to the discussion of the RT-qPCR results. Also, justification of why whole larvae were used instead of specific tissues (such as midgut for oral delivery) is important, since it introduces complications for data analysis and interpretation. Fourth instar S. frugiperda larvae are sufficiently large to do tissue specific dissections with ease. Additional question: were larvae directly homogenized/processed in Trizol, what equipment was used?
Response: Thank you for your comment. In our study, we chose to use whole-body RNA extraction to maintain consistency across the different delivery methods (injection, oral delivery, and spraying). This approach allowed us to directly compare the RNAi efficiency under the same conditions for all methods. While we acknowledge that tissue-specific dissections, such as focusing on the midgut for oral delivery, could provide more detailed insights, our primary goal was to evaluate the overall systemic RNAi response across the entire organism. Regarding the homogenization process, dissected larvae were flash-frozen in liquid nitrogen and then homogenized using a mortar and pestle (Lines 141-142, Page 3). This ensured efficient and consistent disruption of the larval tissues for RNA extraction using TRIzol.
C3: Text description of figure 1C still indicates that no significant difference was found at 48 hrs, despite the graph indicating otherwise. 1D: if representative images are were used according to author response, indicate this.
Response: Thank you for your comment. We have revised Figures 1C and 1D accordingly. In Figure 1C, we have clarified that a significant reduction in expression also occurs at 48 hours post-injection (Lines 296, Page 6). In Figure 1D, we have explicitly stated that the images shown are representative (Lines 314, Page 7).
C4: Figures 1, 2 and 3. How many larvae were used for TRIzol extraction for each sample? If it is a pool of several larvae, differences in efficiency between larvae may also contribute to masking RT-qPCR results. Not all larvae are silenced equally.
Response: Thank you for your comment. In our study, each RNA sample was extracted from a pool of three treated larvae (Lines 143, Page 3). Pooling larvae helps mitigate the effects of individual variability by providing an average representation of the RNAi response across multiple individuals. This approach is particularly useful for detecting overall trends in gene silencing efficiency. We applied the same pooling strategy for all delivery methods (injection, oral delivery, and spraying) to ensure consistency in our comparisons. Finally, analyzing individual larvae would require significantly more resources, including reagents and time, which were limited in our study.
Reviewer 2 Report (Previous Reviewer 2)
Comments and Suggestions for Authors
- line 74 please add coma after "RNA (siRNA)" and add "the" before S. frugiperda.
- Line 94 contains a statement in the purpose: 'Chitosan/dsRNA-V-ATPaseD nanoparticles combined with emamectin benzoate-lufenuron formulation in bioassays.' This suggests that the active substance and chitosan/dsRNA-V-ATPaseD were combined. However, the title implies that the two substances were applied separately. The material and method also includes the use of two active ingredients separately and together. Please review the purpose.
Author Response
C1. line 74 please add coma after "RNA (siRNA)" and add "the" before S. frugiperda.
Response: These changes have been made (Lines 93, Page 2)
C2. Line 94 contains a statement in the purpose: 'Chitosan/dsRNA-V-ATPaseD nanoparticles combined with emamectin benzoate-lufenuron formulation in bioassays.' This suggests that the active substance and chitosan/dsRNA-V-ATPaseD were combined. However, the title implies that the two substances were applied separately. The material and method also includes the use of two active ingredients separately and together. Please review the purpose.
Response: We thank the reviewer for pointing out the ambiguity in the statement regarding the purpose of the study. Indeed, the chitosan/dsRNA-V-ATPaseD nanoparticles were applied first, followed by the emamectin benzoate-lufenuron formulation after a 24-hour interval. This sequential application was designed to assess whether the RNAi-mediated silencing of V-ATPaseD could enhance the efficacy of the chemical insecticides. We acknowledge that the wording in Line 94 could be misinterpreted to imply that the two substances were combined into a single formulation. To avoid confusion, we have revised the purpose statement to clearly reflect the sequential application of the two treatments. The updated text (Lines 111-115, Page 2) now reads: "In this study, we use RNAi to target the V-ATPaseD in S. frugiperda and evaluate the additive effects of chitosan/dsRNA-V-ATPaseD nanoparticles and emamectin benzoate-lufenuron formulation in bioassays, with the dsRNA treatment applied first, followed by the chemical insecticide after 24 hours." We hope this clarification resolves the reviewer’s concern and improves the clarity of our manuscript.
This manuscript is a resubmission of an earlier submission. The following is a list of the peer review reports and author responses from that submission.
Round 1
Reviewer 1 Report
Comments and Suggestions for Authors
Methods:
What is the composition of the artificial diet the larvae were reared on? Please specify or include reference to published protocols.
Although widely used in papers, b- actin does not have stable expression across all conditions or development stages. It may not be the best candidate as a reference gene. A better alternative would have been ribosomal protein genes. As the RT-qPCR experiments have already been run using this reference, an additional experiment checking for stable expression in b-actin in the different experimental conditions in this paper is necessary to consider the expression results valid. Specifically the authors must test larvae exposed to the nanoparticles, lufenuron, and in fourth instar but only on clean artificial diet.
In none of the methods of RNAi administration do the authors specify which tissues were recovered from larvae to perform total RNA extraction, and much less how these tissues were handled until extraction to limit changes in gene expression. Given the fundamental use of gene expression for the entire paper, the omission of these details is not admissible. Please provide all of these details.
What is the pH of the buffers used during chitosan nanoparticle preparation?
Was there consistency in the location of where the nanocomplexes where applied topically to the larvae? Between which segments was this applied? This could influence how much of the solution reaches the desired target tissue.
There are instances of multiple t-tests being performed for a set of data points, but no correction for multitesting is mentioned. This is statistically necessary when these many comparisons are made. Please review your data with multitesting in place where 3 or more comparisons are made in one experimental condition (i.e. several dose treatments at one time point).
Figure 1b: Which condition was used as reference for relative expression? None of the data presented reach 1.0 expression (maybe La-5), so it is unclear to what are all the instars being compared. Specify this.
Figure 1c: if changes in expression at 48 hr are not significant, why is there a ** indicated at the top of the graph?
Figure 1d: The acronym CK is first used in this figure, although it is not specified in the text until later. Please specify here first. The description in the text is not clear from the images. The delay in larvae fed with dsRNA for VATPase is not apparent as they start pupating at 120h, the CK column at 144h only has 2 vs 3 pupae. At 168h the number of individuals presented changes for each column. In methods, it is specified that 10 larvae were used per biological replicate, but in general only 5 are shown. Numeric representation of weight and number of pupae at n hours is a more adequate data. Also, according to methods, length of larvae was not measured, therefore the scale bar is not useful.
Figures 2 and 3. A similar comment here to figure 1. The selected images of larvae do not really reflect the observation described. The reported delay and stunted growth are not very evident in these images. In figure 3, there are more than 10 pupae shown for CK and eGFP. Why? Wasn’t each biological replicate only performed on 10 larvae as indicated in the methods? Compare all of these with the results obtained in figure 4, where the phenotype is clear.
Figure 5: To make it easier to relate text and image, either change the x axis to hours, or refer in the text to days. At 48 and 72 there is a higher mortality with the combined treatment. Please overlay the result of the statistical test of the Mantel Cox test in order to use the “significant difference” wording. The result of this test is not shown or indicated in the text. This comment applies as well to figure 4C However, “synergy” in bioassays is not properly used, since it is defined as an effect above that of either treatment alone. This does not seem to be the case here. There is no curve of only V-ATPaseDi alone to make this conclusion. Perhaps only an additive effect of the transient silencing, as the end point of the bioassay is the same in all treatments, even the non target eGFP dsRNA.
Discussion.
The first paragraph again reiterates a phenotype of delay in development and pupation that is not as clear cut with the data presented in this paper. Most of the data in figure 2 and 3 show no difference in weight, it is only significantly lower in one data point,and this could be because of lack of multitesting corrections. The conclusion stated at the start of the second paragraph is grossly overreaching the data in this paper. It should be eliminated.
To repeat a previous comment, there is not enough data to speak of synergy between dsRNA and the lufenuron, perhaps only additive. No synergy factor was calculated at all. Any instance of the word “synergy”, including the title of the manuscript, must be removed and replaced with other language. In general, the authors need to tone down the conclusions presented in the paper, as the data is not strong enough for the assertions in the manuscript.
Reviewer 2 Report
Comments and Suggestions for Authors
-Abstract should be rearranged, and the methods should be summarized briefly. The explanation of the results is a bit complicated and should be simplified. The abstract and material-method sections are not compatible; the times mentioned in the abstract are not specified in the material-method section. These should also be specified in the material-method section.
-The author name should be added after the scientific name in the title. The author name should be removed after the scientific name in the first sentence of the introduction.
-In the introduction, the sentence 'and genetically engineered Bt crops expressing Bacillus thuringiensis (Bt) toxins' is referred to as biological control. This is an incorrect statement. Genetically engineered crops are not considered part of biological control. The relevant sentence should be edited.
-The terms eGFP gene and eGFP-dsRNA appear in section 2.4. However, they do not appear anywhere before. What is eGFP-dsRNA used for? It should be explained.
-In section 2.6, the reference writing should be corrected in accordance with the rules. The author and year should be deleted.
-In data analysis, how larval measurements were made, during which periods they were made, and the survival data should also be explained.
-The number of repetitions should be specified when determining LC50 and LC30. What are the criteria for dose selection? It should be specified in which mortality range the selected doses were distributed
-Why was the sublethal effect evaluated only in the larval stage? Why was it not continued in the adult stage? Fecundity is important data for the sublethal effect
